# Oxidative-Stress-Associated Proteostasis Disturbances and Increased DNA Damage in the Hippocampal Granule Cells of the Ts65Dn Model of Down Syndrome

**DOI:** 10.3390/antiox11122438

**Published:** 2022-12-09

**Authors:** Alba Puente-Bedia, María T. Berciano, Carmen Martínez-Cué, Miguel Lafarga, Noemí Rueda

**Affiliations:** 1Departamento de Fisiología y Farmacología, Facultad de Medicina, Universidad de Cantabria, 39011 Santander, Spain; 2Departamento de Biología Molecular, “Red Sobre Enfermedades Neurodegenerativas (CIBERNED)”, Universidad de Cantabria-IDIVAL, 39011 Santander, Spain; 3Departamento de Anatomía y Biología Celular, “Red Sobre Enfermedades Neurodegenerativas (CIBERNED)”, Universidad de Cantabria-IDIVAL, 39011 Santander, Spain

**Keywords:** Ts65Dn, hippocampus, down syndrome, granule cells, oxidative stress, DNA damage, proteostasis

## Abstract

Oxidative stress (OS) is one of the neuropathological mechanisms responsible for the deficits in cognition and neuronal function in Down syndrome (DS). The Ts65Dn (TS) mouse replicates multiple DS phenotypes including hippocampal-dependent learning and memory deficits and similar brain oxidative status. To better understand the hippocampal oxidative profile in the adult TS mouse, we analyzed cellular OS-associated alterations in hippocampal granule cells (GCs), a neuronal population that plays an important role in memory formation and that is particularly affected in DS. For this purpose, we used biochemical, molecular, immunohistochemical, and electron microscopy techniques. Our results indicate that TS GCs show important OS-associated alterations in the systems essential for neuronal homeostasis: DNA damage response and proteostasis, particularly of the proteasome and lysosomal system. Specifically, TS GCs showed: (i) increased DNA damage, (ii) reorganization of nuclear proteolytic factories accompanied by a decline in proteasome activity and cytoplasmic aggregation of ubiquitinated proteins, (iii) formation of lysosomal-related structures containing lipid droplets of cytotoxic peroxidation products, and (iv) mitochondrial ultrastructural defects. These alterations could be implicated in enhanced cellular senescence, accelerated aging and neurodegeneration, and the early development of Alzheimer’s disease neuropathology present in TS mice and the DS population.

## 1. Introduction

Down syndrome (DS) is the most common genetic cause of intellectual disability [1] produced by a complete or partial triplication of the human chromosome 21 (Hsa21). Although a variety of cognitive domains are compromised in DS, hippocampal-dependent learning and memory are particularly affected because of anatomical and functional alterations present from prenatal stages [2,3]. These brain alterations are aggravated during later life stages, leading to early aging, increased neurodegeneration, further loss of cognitive abilities, and the development of Alzheimer’s disease (AD) neuropathology [4,5].

One of the main neuropathological processes responsible for the cognitive alterations and deficits in neuronal function in DS is the enhancement of brain oxidative stress (OS). This alteration is partially due to the overexpression of several Hsa21 genes that encode specific proteins directly or indirectly involved in the production of reactive oxygen species (ROS) [6,7,8]. One of the triplicated genes that causes increased OS in the DS brain is *SOD1*, which encodes for the enzyme superoxide dismutase (SOD1). This enzyme converts superoxide anions into molecular oxygen and hydrogen peroxide (H_2_O_2_). Its overexpression results in the formation of an excessive amount of H_2_O_2_ that is not adequately metabolized by the activity of the antioxidant enzymes catalase (CAT) and glutathione peroxidase (GPx), leading to its accumulation in the cytosol and to the formation of ROS in DS. In addition to *SOD1* overexpression, alterations in other components of the endogenous antioxidant system, such as low levels of antioxidant enzymes and failures in antioxidant responses, further contribute to the accumulation of ROS in DS [9,10,11,12].

Mitochondria are the main source and one of the main targets of free radicals. In DS, numerous alterations in the structure and function of mitochondria have been demonstrated [13,14,15,16]. First, there is compelling evidence of alterations in the mitochondrial oxidative phosphorylation system in different cell types and tissues from prenatal stages to advanced age in DS [17]. In addition, DS cells show a decline in mitochondrial biogenesis and turnover [13,18] and defective mitophagic activity [14,15]. These defects lead to the accumulation of damaged mitochondria, increased mitochondrial mass, and augmentation of OS [14]. Increased OS in DS causes mitochondrial dysfunction, favoring the production of ROS and altering energy metabolism with reduced ATP generation [15,16,17].

The augmentation of ROS production promotes damage to proteins, lipids, and DNA. Thus, neurons from DS individuals present high levels of different markers of lipid peroxidation, oxidized proteins, and DNA damage [19,20,21]. In DS, oxidative modifications of lipids and proteins cause structural and functional damage to membrane-bound organelles affecting the intracellular degradative machinery [22,23,24]. Finally, in DS cells, OS induces DNA damage and defective DNA repair, which can produce genomic instability and, ultimately, activation of apoptosis [25,26,27,28].

In the DS brain, chronic exposure to OS and failures in the production of ATP disrupt the proper function of two main intracellular degradation systems: autophagy and the ubiquitin-proteasome system (UPS) [24,29]. Indeed, reduced proteasome activity and accumulation of polyubiquitinated proteins have been observed in the DS brain and in fibroblasts from DS individuals [24,30,31]. Similarly, in DS cells, defective autophagy associated with the upregulation of the mTOR pathway has been demonstrated [14,23]. Primary fibroblasts from individuals with DS also exhibit lysosomal deficits such as decreased lysosomal turnover of autophagic protein substrates, impaired lysosomal acidification, and altered lysosomal enzymatic activity [32]. These perturbations lead to the early accumulation of unwanted proteins and to a lack of clearance of toxic aggregates or defective organelles that play a role in the characteristic early neurodegeneration found in DS.

Thus, increased OS in the DS brain is an early event that negatively affects brain development, but, during later life stages, OS in DS is further aggravated [23,33]. Persistent oxidative damage to biomolecules and cell organelles, including components of the intracellular degradative systems, are responsible for multiple neuronal alterations, resulting in accelerated brain aging and neurodegeneration, which contribute to the early appearance of AD neuropathology and cognitive decline [6,22,34,35].

To understand the neurobiology and pathophysiology that characterize DS, a variety of mouse models have been created [2]. Among them, the partial trisomic Ts65Dn (TS) mouse replicates numerous DS phenotypes, including hippocampal-dependent learning and memory deficits [36]. These cognitive disabilities of the TS mouse have been attributed to hippocampal hypocellularity from defective neurogenesis, as well as to morphological, electrophysiological, and functional anomalies of numerous neuronal populations, including the granule cells (GCs) of the dentate gyrus [7,37,38,39,40,41,42,43]. Similar to DS, in the TS mouse, these alterations are aggravated throughout life by a variety of neuropathological processes, including increased OS [37,38,39]. In fact, TS mice exhibit an oxidative and mitochondrial brain dysfunction profile similar to the one previously described in humans with DS [36,37,38,44,45,46,47,48]. The increase in OS in TS mice has also been related to (i) impairments of the UPS and autophagy [27,49], (ii) the high density of cells with a senescent phenotype [35,39,46], and (iii) the premature neurodegeneration that occurs in the brain of this model starting at the age of 6 months [44,48,50].

To better understand the hippocampal oxidative profile in the TS mouse model of DS, this study aimed to analyze the effect of increased OS in GCs of the dentate gyrus on two main systems for neuronal homeostasis: the DNA damage response (DDR) and proteostasis, as well as their potential relationships with the epigenetic changes previously reported in this neuronal population [41]. In particular, we focused on (i) the nuclear profile of DNA damage foci and their repair mediated by the replication-independent upregulation of γH2AX gene expression, (ii) the reorganization of nuclear proteolytic factories of the UPS (“clastosomes”), which are associated with the decline in proteasome activity, (iii) the abnormal cytoplasmic aggregation of ubiquitinated proteins, presumably by a liquid-to-liquid phase separation mechanism, and (iv) the ultrastructural reorganization of lysosomes with the formation of lysosomal-related structures containing lipid droplets of cytotoxic peroxidation products. Collectively, the dysfunction of these neuronal processes in the GCs of 6-month-old TS mice must represent an essential contribution to the neuromorphological and functional alterations found in the early stages or to the accelerated aging and early neuropathological manifestations of AD that occur during adulthood in TS mice.

## 2. Materials and Methods

### 2.1. Animals

This study was approved by the Ethics Committee of the University of Cantabria and all the experiments were conducted in accordance with the Declaration of Helsinki and the European Communities Council Directive (86/609/EEC of 24 November 1986). TS mice were generated by repeated backcrossing of B6EiC3Sn a/A-Ts(17 < 16>)65Dn females with C57BL/6Ei × C3H/HeSNJ (B6EiCSn) F1 hybrid males. The parental mouse generation was provided by Robertsonian Chromosome Resources (The Jackson Laboratory, Bar Harbor, ME, USA), and mating was performed in the University of Cantabria animal facilities. The animals were karyotyped using quantitative real-time PCR (RT-qPCR) as previously described by García-Cerro et al. [51]. In all the experiments, 6-month-old (±15 days) TS mice were compared with their euploid control littermates.

### 2.2. Immunofluorescence and Confocal Microscopy

To analyze nuclear and cytoplasmatic histological alterations by immunofluorescence, at least 3 animals of each karyotype were euthanized. Mice were anesthetized and perfused with paraformaldehyde 3.7% in phosphate buffered saline (PBS), and their brains were removed and post-fixed for 1 h at room temperature and coronally sliced (200 μm thick) in a vibratome. Isolated removal of the granular cell layer (GCL) of the dentate gyrus was performed following Palkovits’s procedure [52], but using razor blades for the microdissection of the GCL under a stereomicroscope. Small tissue fragments were processed for the mechanical dissociation of GC bodies. GC dissociates were obtained in squash preparations following the protocol described by Pena et al. [53]. Briefly, microdissected fragments of the GCL were transferred to a drop of PBS on a siliconized slide (Super-Frost-Plus, Menzel-Gläser, Germany). A coverslip was then located on the top of the slide and the hippocampal tissue was squashed by mechanical percussion with a histologic needle to dissociate GC bodies. The preparation was then frozen in dry ice and the coverslip was removed with a razor blade. Thanks to this protocol, most GC bodies remained adhered to the slide.

For the immunofluorescence studies, squash preparations were sequentially treated with 0.1 M glycine in PBS, 0.5% TritonX-100 in PBS, and incubated with the primary antibodies: monoclonal mouse anti-phospho-histone H2A.X (Ser139) (1:100; 05-636, Millipore, Darmstadt, Germany), mouse anti-ubiquitin (1:100; sc-8017, Santa Cruz Biotechnology, Santa Cruz, CA, USA), polyclonal rabbit anti-PSMB5 (1:100; PA1-977, Invitrogen, Vilnius, Lithuania), and polyclonal goat anti-cathepsin D (1:100; sc-6486, Santa Cruz) overnight at 4 °C. Then, the samples were washed with PBS with 0.05% Tween-20 and incubated for 45 min with the specific secondary antibody conjugated with FITC or TexasRed (Jackson Laboratories, USA Jackson, West Grove, PA, USA). Some samples were counterstained with propidium iodide for 15 min at room temperature. Finally, the preparations were mounted with Vectashield with DAPI^®^ (Vector Laboratories, Inc., Burlingame, CA, USA) and confocal images were obtained with an LSM510 laser confocal microscope (Zeiss, Oberkochen, Germany) using 63× oil (1.4 NA) objective.

To determine lipofuscin autofluorescence, the squash preparations were processed as previously described and directly mounted with Vectashield with DAPI^®^. To obtain the images, samples were excited with an argon laser (488 nm) in an LSM510 confocal microscope.

Quantitative analyses were performed using the ImageJ software (U.S. National Institutes of Health, http://imagej.nih.gov/ij). Quantification of the number of foci of DNA damage and of cathepsin D per cell, the number of clastosomes and inclusions per cell, and the percentage of cells with DNA foci and with inclusions were performed in at least 100 GCs per animal.

### 2.3. Conventional Electron Microscopy

To analyze the ultrastructure of the lysosomes and mitochondria of GCs, 3 animals of each karyotype were perfused with 3% glutaraldehyde in 0.1 M phosphate buffer pH 7.4 under deep anesthesia. Brains were removed and post-fixed overnight with the perfusion buffer. Then, 300 μm thick slices were obtained using a vibratome and small fragments of the GCL of the hippocampus were dissected. Tissue fragments were post-fixed with 2% osmium tetroxide, dehydrated in increasing concentrations of ethanol, and embedded in Araldite. Ultrathin sections mounted in copper grids were stained with lead citrate and uranyl acetate and examined using a JEOL 201 (Tokyo, Japan) electron microscope operated at 80 kV. Electron micrographs were recorded using a camera (Orius 1200A; Gatan, Pleasanton, CA, USA).

### 2.4. Detection of ROS Levels

ROS production was assessed on 6 hippocampi from each group of animals using a fluorometric assay, as described previously [44,54]. This method is based on the oxidation of the nonfluorescent 2′,7′-dichlorodihydrofluorescein diacetate (H2DCFDA), in the presence of ROS such as hydrogen peroxide, to the highly fluorescent 2′,7′-dichlorofluorescein (DCF). Hippocampi were weighted and sonicated in 100 µL 40 mM Tris–HCl buffer (pH 7.4) on ice. Next, 50 µL of the hippocampal homogenates were incubated with H2DCFDA (5 μM final concentration; Molecular Probes, Eugene, OR, USA) at 37 °C for 30 min. Fluorescence was monitored on a Synergy HTX BioTek Multimode plate reader (Santa Clara, CA, United States) (λexcitation = 488 nm; λemission = 525 nm). Quantification of ROS was assessed from a standard curve of DCF (Sigma, St. Louis, MO, USA) in ethanol (range 1 μM–0.01 μM). Values were normalized by protein weight, and the results were expressed as μM DCF produced per mg protein.

### 2.5. In Situ Determination of Oxidized Lipids

To prepare the GC squashes for the determination of oxidized lipids, small fragments of GCL were dissected, cells were dissociated and frozen in dry ice, and samples were air-dried, washed in PBS, and stored at 4 °C. The samples were incubated with the probe C11-Bodipy581/591 at a concentration of 1 μg/μL for 30 min at 37 °C to detect lipid peroxidation (Molecular Probes, Madrid, Spain). Then, samples were washed in PBS and mounted with Vectashield-DAPI^®^ (Vector Laboratories, Newark, CA, USA), and images were obtained immediately using a Zeiss LSM 520 microscope.

C11-Bodipy581/591 fluorescence intensity was quantified, delimiting three circles within the cytoplasm of each cell to perform a raw intensity analysis with the ImageJ software. Quantifications of fluorescence intensity and of the number of Bodipy foci were performed in at least 100 GCs from three mice of each karyotype.

### 2.6. Determination of Protein Oxidation: Protein Carbonyl Content

The oxidation of proteins results in the production of stable carbonyl groups, which can be used as a measure of oxidative injury. Hippocampal carbonyl content was determined on 6 hippocampi from each group of animals using a Protein Carbonyl Content Assay Kit (Sigma-Aldrich; Darmstadt, Germany, Ref: MAK094). Hippocampal samples were homogenized and processed following the manufacturer’s instructions. On each sample, protein concentration was estimated using the BCA Protein Assay Kit (Pierce, Rockford, IL, USA). Analyses were always performed in duplicate. OD_375_ values were detected on a microplate reader (Synergy HTX multimode plate reader, BioTek). The carbonyl content of each sample (expressed as nmol carbonyl/mg protein) was calculated according to the manufacturer’s instructions.

### 2.7. Real-Time Quantitative PCR

Six hippocampi of TS and CO mice were used for the qRT-PCR studies. The animals were decapitated after being anesthetized and the hippocampi were quickly removed, frozen, and stored at −80 °C. RNA was isolated using RNAeasy Plus Mini Kit (Qiagen, Hilden, Germany) and reverse-transcribed to cDNA using a Revert Aid H Minus First Strand cDNA Synthesis Kit (Thermo Fisher, Vilnius, Lithuania). The cDNA concentration was measured in a spectrophotometer (Nanodrop Technologies ND-1000; Wilmington, DE, USA) and adjusted to 1 μg/μL. The expression of the mRNA candidates was analyzed by RT-qPCR using gene-specific SYBR Green primers. The threshold cycle (Ct) for each sample was determined with three technical triplicates, and the results were normalized to the housekeeping gene GAPDH. Relative expression was calculated according to the 2-(ΔΔCt) equation [55]. The SYBR Green-based specific primers for murine RNAs were as follows: for Nfr2 5′-ccgctacaccgactacgatt-3′ and 5′-tgtttcctgttctgttccccg-3′, for H2AX 5′-tcctgcccacactccagg-3′ and 5′-tcagtactcctgagaggcctgc-3′, for H2AX-3′end 5′-tccccacacctccacaaag-3′ and 5′-ggaaagagaaaggatgggggacg-3′, for Pink1 5′-cgacaacatccttgtggagtgg-3′ and 5′-cattgccaccacgctctacact-3′, for Prkn 5′-ccagaggaaagtcacctgcgaa-3′ and 5-gttcgagcagtgagtcgcaatc-3′, for Gpx1 5′-cgctctttaccttccygcggaa-3′ and 5′-agttccaggcaatgtcgttgcg-3′, for Cat 5′-tgcagatacctgtgaactgtcc-3′ and 5′-agctgttggggtaatagttggg-3′, for Hmox1 5′-cactctggagatgacacctgag-3′ and 5′-gtgttcctctgtcagcctcacc-3′, for Nqo1 5′-gccgaacacaagaagctggaag-3′ and 5′-ggcaaatcctgctacgagcact-3′ and for Gapdh 5′-aggtcggtgtgaacggatttg-3′ and 5′-tgtagaccatgtagttgaggtca-3′.

### 2.8. Western Blotting

For western blot analysis, 6 mice of each karyotype were euthanized by decapitation. Their hippocampi were dissected, homogenized, and lysed as previously reported by García-Cerro et al. [51]. The homogenates were boiled for 10 min and sonicated for 5 cycles of 30 s on/off at 4 °C using a Bioruptor Plus (Diagenode, Valencia, Spain) and left on ice for 20 min. The protein in the supernatants of each sample was quantified following the protocol described by Lowry et al. [56]. The same amounts of protein were subjected to SDS-PAGE electrophoresis and then transferred to polyvinylidene difluoride (PVDF) membranes. Blots were incubated with the following primary antibodies: polyclonal anti-Cu/Zn SOD (ADI-SOD, Enzo Life Sciences, Barcelona, Spain), monoclonal mouse anti-ubiquityl-histone H2A (05-678, Millipore), polyclonal rabbit anti-Proteasome 20S (PW8155, Enzo Life Science), monoclonal mouse anti-Proteasome 19S Rpt5S6a subunit (PW8770, Enzo Life Science), and polyclonal goat anti-cathepsin D (sc-6486, Santa Cruz) overnight at 4 °C, extensively washed, and exposed to a specific secondary antibody donkey anti-goat IRDye800CW, goat anti-rabbit IRDye680RD, or goat anti-mouse IRDye800CW (1:10.000; LI-COR Biotechnology, Lincoln, NE, USA) for 45 min at RT. GAPDH (1:2000; sc-32233, Santa Cruz) was used as a loading control. Bands were visualized with LI-COR Odyssey IR Imaging system V3.0 (LI-COR Biotechnology) and quantified with the ImageJ 1.52p software as previously described by García-Cerro et al. [51].

Each individual sample was evaluated in at least three independent experiments.

### 2.9. Proteasome Activity Assay

Chymotrypsin-like activity was assayed in control and TS hippocampal samples following the protocols of Keller et al. [57] and Casafont et al. [58]. The hippocampi of 6 animals from each experimental group were dissected, immediately frozen in dry ice, and stored at −80 °C. Samples were homogenized in lysis buffer containing 10 mM Tris-HCl (pH 7.6), 1 mM EDTA, 4 mM DTT, 2 mM ATP, and 20% glycerol, sonicated on ice and centrifuged at 13,000× *g* for 10 min at 4 °C. The supernatant fraction was collected and the total protein content of each sample was determined using the method of Lowry et al. [56]. In a 96-well tissue culture plate a total of 250 µg of protein from each sample was loaded per well at a concentration of 10 µg/µL with 25 µL of 2× proteasome assay buffer containing 50 mM Tris-HCl (pH 8.8), 0.5 mM EDTA, and 40 µM of the chymotrypsin fluorogenic substrate N-succinyl-Leu-Leu-Val-Tyr-7amido 4-methylcoumarin (Sigma, S-6510). Standards were incubated for 1 h at 37 °C and the reaction was stopped by the addition of ice-cold dH2O. To determine the proteasome chymotrypsin proteolytic activity, the fluorescence was immediately measured on a Synergy HTX BioTek Multimode plate reader (λexcitation = 355 nm; λemission = 460 nm). To ensure that changes in fluorescence were due to the proteasome activity, homogenized samples were also incubated in the same plate with MG132 (Sigma, C-2211), a potent proteasome inhibitor, to a final concentration of 60 µm.

### 2.10. Statistical Analysis

For comparison between TS and CO samples, data were analyzed using SPSS (version 22.0 for Windows, Chicago, IL, USA) and an unpaired Student’s *t*-test. Significance was established at *p* <0.05.

## 3. Results

### 3.1. Increased OS in the Hippocampi of TS Mice Is Associated with Increased ROS Generation, Altered Antioxidant Response, Oxidative Damage to Lipids and Proteins, and Mitochondrial Anomalies in TS GCs

OS reported in DS results from an imbalance between ROS production and the antioxidant system response. Previous studies from our and other laboratories have demonstrated that the hippocampus of the TS mouse is particularly vulnerable to persistent OS [38,46,48]. To better understand the hippocampal OS status of adult TS mice, first, we evaluated ROS production in both groups of animals using an H2DCFDA assay on hippocampal homogenates. The results showed that ROS content was significantly higher in the hippocampi of TS mice compared to the euploid ones (Figure 1A).

Next, we evaluated the expression levels of some components of the antioxidant system. In particular, we analyzed the expression levels of the SOD1 protein, which catalyzes the conversion of O^2−^ to molecular O_2_ and H_2_O_2_. As expected, consistent with the triplication of the *Sod1* gene, Western blotting of hippocampal lysates revealed a significant increase in the levels of SOD1 in TS mice compared to control animals (Figure 1B). As CAT and GPx metabolize the H_2_O_2_ produced by SOD1 into H_2_O and O_2_, we also examined the mRNA expression levels of these antioxidant enzymes. As shown in Figure 1C,D, both groups of animals show similar hippocampal levels of both enzymes. Collectively, in the hippocampus of TS mice, the increment of SOD1 expression is not accompanied by a parallel increase in mRNA expression of CAT and GPx, which may lead to the accumulation of disproportionate levels of H_2_O_2_, causing persistent OS.

Another key element of the antioxidant response is Nrf2, a transcription factor that under high levels of OS binds to antioxidant response elements (AREs) activating the expression of stress response genes, including those of antioxidant enzymes such as NADPH quinone oxidoreductase 1 (NQO1) and heme oxygenase 1 (HO-1). To evaluate the Nrf2 antioxidant response, we analyzed the mRNA expression levels of Nrf2, HO-1, and NQO1 in the hippocampi of TS and CO mice. TS mice displayed decreased levels of *Nrf2* in the hippocampi as compared to CO mice (Figure 1E). Accordingly, the expression levels of genes encoding, NQO1 and HO-1, were also decreased in the hippocampi of TS mice, as compared to CO animals (Figure 1F,G). These results suggest that this antioxidant response may not operate properly, contributing to the exacerbation of OS in the hippocampi of TS mice.

ROS can induce damage to DNA, lipids, and proteins. Thus, in TS GCs, we assessed the detection of peroxidized lipids, a cellular indicator of ROS imbalance and of oxidative damage to lipids [59,60]. For this purpose, we used an in situ assay with Bodipy C11, a lipid-soluble ratiometric fluorescent indicator for the peroxidation of certain lipids of membranous organelles [61]. We found that the cytoplasmic fluorescent signal of Bodipy C11 increased significantly in TS GCs as compared to those of controls (Figure 1H–J). Moreover, TS GCs displayed a higher number of Bodipy C11-positive cytoplasmic bodies (Figure 1K), which presumably correspond to lipid droplets containing peroxidized lipids. To evaluate the protein damage induced by ROS, we measured the carbonyl content of the hippocampal homogenates of both groups of animals. As shown in Figure 1L, TS mice exhibited a non-significant increase in their hippocampal carbonyl content as compared to euploid animals, indicating higher levels of oxidized proteins in this structure.

Mitochondria are also one of the main targets of free radicals and are the major site of ROS production through the oxidative phosphorylation system. In DS and in TS mice, structural and functional alterations are widely associated with high ROS production [14,15,17,47,54]. Therefore, we analyzed whether in TS GCs there was a loss of mitochondrial organization and integrity. Whereas small round or oval mitochondria with a preserved internal structure were the predominant phenotypes in control GC perikarya (Figure 2A), very elongated and polymorphic mitochondria were frequently observed in cell bodies from TS GCs (Figure 2B–D), suggesting an alteration of the mitochondrial network architecture. Moreover, some mitochondria exhibited dilatation of matrix or focal vacuolization with disruption of cristae (Figure 2E,F), structural alterations that were consistent with the OS-induced mitochondrial dysfunction reported in cultured hippocampal neurons from TS mice [62]. This prompted us to investigate whether the presence of damaged mitochondria activates a neuroprotective mechanism for their elimination. We assessed the expression of gene-encoding Pink1 (PTEN-induced putative kinase 1), a Ser/Thr protein kinase that accumulates on the cytosolic surface of dysfunctional mitochondria to induce, together with Parkin, their degradation via mitophagy [63]. We found a significant increase in Pink1 mRNA in hippocampal RNA extracts from TS mice as compared with those of controls (Figure 2G). Despite the upregulation of Pink1 in TS mice, no changes were observed in the hippocampal levels of Parkin mRNA between animals of both genotypes (Figure 2H).

### 3.2. Oxidative Stress in TS GCs Induces DNA Damage

Oxidative damage to DNA is emerging as an early cellular event in neurodegeneration. Increased ROS production causes DNA damage by increasing the generation of double strand breaks (DSBs), the most harmful form of DNA lesions, and altering the DNA repair mechanisms [64,65].

To determine whether the enhanced OS induces oxidative DNA damage in 6-month-old TS GCs, we performed immunofluorescence for the phosphorylated histone H2AX (named γH2AX), a post-translational modification (PTM) used as a marker of DSBs [66,67]. Dissociated control and TS GCs from the dentate gyrus were immunolabeled for γH2AX and counterstained with propidium iodide, a cytochemical marker of RNA in neurons [68]. Typical nuclear DNA damage foci immunoreactive for γH2AX were found in some GC nuclei (Figure 3A,B). They appeared as sharply defined nuclear mini compartments enriched in γH2AX (Figure 3B inset) and presumably assembled by a mechanism of liquid-to-liquid phase separation of DNA damage signaling and repair components at sites where DSBs concentrate [69,70]. The quantitative analysis of the proportion of GCs containing γH2AX-positive foci in whole nuclei revealed a significant increase in TS GCs as compared with age-matched control neurons (55.4 vs. 24.6%, *p* < 0.05) (Figure 3C). Accordingly, the mean number of these foci per nucleus was also significantly increased in TS GCs relative to control neurons (0.98 vs. 0.36%, *p* < 0.01) (Figure 3D).

Given the well-established role of the histone H2AX in the cellular response to genotoxic stress [71], we analyzed whether DNA damage in TS GCs induced changes in the expression level of its mRNA. H2AX is a nucleosomal histone variant with a 3′-end characteristic of replicating-histone mRNAs that undergo specific 3′-end cleavage, the only pre-mRNA processing step [72]. The qRT-PCR performed in hippocampal RNA extracts revealed a significant increase in H2AX mRNA in TS mice compared with controls (Figure 3E). Next, we investigated the possible existence of defective H2AX mRNA processing in hippocampal GCs of TS mice. We quantified the levels of 3′-extended H2AX transcripts that might accumulate from impaired processing [73]. The estimation by qRT-PCR of unprocessed H2AX mRNA revealed a slight, but non-significant, increase in 3′-extended H2AX transcripts in TS hippocampi relative to controls (Figure 3F).

In addition to the phosphorylation of H2AX, the ubiquitination of H2AX (Ub-H2AX) plays an important role in regulating DNA damage response (DDR) and chromatin remodeling [74,75,76]. Therefore, we investigated whether the increased DNA damage in TS GCs correlated with augmented expression of this H2AX PTM. As expected, Western blotting of hippocampal lysates showed a significant increase in Ub-H2AX in TS mice as compared with controls (Figure 3G), suggesting the active role of H2AX ubiquitination in the DDR of GCs.

Collectively, our results supported that the aging-dependent DNA damage is accelerated in GCs from the TS mouse model of DS, possibly owing to increased cellular OS.

### 3.3. Increased OS Is Associated with Proteostasis Disturbances in TS GCs

Studies in DS also indicated that excessive ROS formation can target intracellular quality control mechanisms such as the UPS and autophagy-lysosomal system, which are closely interconnected, promoting the formation of protein aggregates or the accumulation of unwanted toxic products [24,29].

#### 3.3.1. Dysfunction of the Proteasome

Protein clearance via the UPS requires the polyubiquitination of target proteins, mediated by the sequential activity of the ubiquitin-activating enzyme (E1), ubiquitin-conjugating enzyme (E2), and ubiquitin ligase (E3), and subsequent recognition of polyubiquitinated proteins by the 19S regulatory proteasome followed by their degradation by the catalytic proteasome 20S [77]. Alterations and reduced functionality of the proteasome have been reported in the human frontal cortex, cultured cell lines from DS individuals, and the cerebella of TS mice [24,27,30].

On these bases, we therefore investigated whether dysfunction of the proteasome affects the response of TS GCs to OS. We performed immunofluorescence for the catalytic 20S proteasome on dissociated GCs from control and TS mice hippocampi. In the GCs of both genotypes, preferential nuclear staining, excluding the nucleolus, was observed (Figure 4A,B). Importantly, control GCs frequently exhibited nuclear foci with a high concentration of 20S proteasomes (Figure 4A inset) that corresponded to “clastosomes”, nuclear proteolytic factories of the UPS [78,79,80,81]. The quantitative analysis in whole nuclei of the mean number of clastosomes per GC nucleus revealed a significant increase in control GCs as compared with TS neurons (Figure 4E). Given that this finding was consistent with a reduction of proteasome activity in TS GCs, we next performed a proteasome activity assay based on the cleavage of LLVY-AMC for chymotrypsin-like activity, a major proteolytic component of the catalytic 20S proteasome [58,82]. As expected, there was a significant decrease in proteasome activity in hippocampal lysates from TS mice (approximately 35%) compared to controls (Figure 4H). Moreover, Western blot determination of protein levels of both regulatory 19S and catalytic 20S subunits in hippocampal lysates did not show changes when TS and control mice were compared (Figure 4I), suggesting that the reduction of proteasome activity was not due to changes in the expression of its subunits.

Given that the proteolysis via the UPS is mostly, but not exclusively, of polyubiquitinated proteins, we investigated their subcellular distribution using an antibody that recognizes ubiquitin-protein complexes (Ub-proteins). As illustrated in Figure 4C,D, a diffuse fluorescent signal of Ub-proteins was observed in the nucleus and cytoplasm in both control and TS GCs. Importantly, we found nuclear foci of higher signal intensity, which stood out against the diffuse nuclear fluorescence, particularly in control GCs (Figure 4C, inset). Moreover, bright and sharply defined round aggregates or inclusions of Ub-proteins were found in the cytoplasm of TS GCs (Figure 4D, inset). The quantitative analysis of both the proportion of GCs with cytoplasmic aggregates and the mean number per neuron revealed a significantly higher number of TS GCs as compared with control neurons (Figure 4F,G).

#### 3.3.2. Dysfunction of Lysosomal System

Based on current literature, an early neuropathological manifestation in DS is the dysfunction of endosomal protein sorting and trafficking, which ultimately leads to the accumulation of abnormal proteins and impairment of lysosomal functions [32,83,84]. Studies in the DS brain have suggested that increased oxidation of components of the autophagic–lysosomal system alters its functionality, exacerbates the inhibition of this degradative mechanism, and favors proteostasis network dysfunction [29].

To evaluate the early lysosomal response to the OS and lipid peroxidation in TS GCs we studied the expression of cathepsin D, a molecular marker of lysosomes, as well as the ultrastructural features of lysosome-related cytoplasmic bodies. Cathepsin D-positive bodies were clearly identified by immunofluorescence in both control and TS GCs (Figure 5A,B). While most of the cathepsin D-positive bodies were large and polymorphic in TS GCs, smaller and round bodies were predominant in control neurons at 6 months of age (Figure 5A,B). The quantitative analysis of the average number of cathepsin D-positive lysosomes in the whole neuronal body revealed a significant increase in TS GCs compared with control neurons (Figure 5C). However, Western blot determination of cathepsin D in hippocampal lysates did not show significant changes in its protein levels when control and TS samples were compared (Figure 5D). This finding suggested an abnormal accumulation of dysfunctional lysosome-related structures in TS GCs without a parallel increase in cathepsin D-dependent protein degradation.

To investigate this hypothesis, we analyzed the ultrastructural phenotype of lysosomal bodies, a parameter linked to their functional activity [85]. At 6 months of age, most lysosome-related structures in control GCs consisted of round and homogeneous electron-dense bodies, identified as primary lysosomes, and more heterogeneous secondary lysosomes containing small electron-dense granules of protein condensates (Figure 6A). In contrast, TS GCs commonly displayed larger, polymorphic lysosome-related bodies with a heterogeneous internal structure (Figure 6B–E) that sometimes form clusters (Figure 6B–E). Intralysosomal components included (i) round electron-lucent domains identified as lipid droplets, (ii) electron-dense granules of variable number and size of protein aggregates, and (iii) membranous-derived structures (Figure 6B–E). A fraction of the intralysosomal lipid droplets found in TS GCs seemed to correspond to the storage of peroxidized lipids stained with Bodipy C11 (Figure 1I). Moreover, some complex bodies with several lipid droplets may be lipofuscin residual granules containing indigestible oxidized lipids and proteins. Indeed, some cytoplasmic bodies in TS GCs displayed a strong autofluorescence with confocal microscopy (Figure 6F), a typical characteristic of lipofuscin granules [61,86].

Collectively, the early onset accumulation in TS GCs of abnormal lysosome-related bodies enriched in peroxidized lipids was consistent with a lysosomal dysfunction that contributes to proteostasis disturbance in this neuronal population.

## 4. Discussion

Our results indicate that the hippocampus of the TS mouse exhibited altered redox homeostasis and the GCs of the dentate gyrus showed alterations associated with increased OS in two systems essential for neuronal homeostasis: DDR and proteostasis, particularly of the UPS and lysosomal systems. In summary, GCs of TS mice displayed: (i) increased DNA damage, (ii) reorganization of nuclear proteolytic factories accompanied by a decline in proteasome activity and cytoplasmic aggregation of ubiquitinated proteins, (iii) reorganization of lysosomes with the formation of lysosomal-related structures containing lipid droplets of cytotoxic peroxidation products, and (iv) mitochondrial ultrastructural defects.

It is well established that GCs of the dentate gyrus play a fundamental role in memory formation since they are the first element of the hippocampal trisynaptic circuit [87]. Therefore, the OS-dependent detrimental effects of increased DNA damage and proteostasis disturbances on TS GCs reported here could compromise proper neuronal functions in young animals and promote neurodegeneration and the development of AD neuropathology in aged mice. Consequently, these effects can contribute to the hippocampal-dependent learning and memory deficits that are present in this model of DS.

Excessive OS in DS is partly due to the triplication of several genes that are directly or indirectly implicated in the production of an excess of free radicals and in the dysfunction of the antioxidant defense system, creating a redox imbalance and the accumulation of oxidative damage [6,7,29]. In particular, in the TS mouse, increased ROS production has been demonstrated in different areas of the brain and at different ages (including neonatal stages) [44,54,88]. Regarding the antioxidant defense system, in the hippocampus of this model, higher SOD1 activity without a concomitant increase in complementary antioxidant GPx and CAT enzymatic activities has been previously found at 5 and 12 months of age [38,46]. Similarly, in this study, 6-month-old TS mice display increased ROS content in their hippocampi. In addition, consistent with the extra copy of the *Sod1* gene, TS mice show an augmentation in the expression of SOD1, but without a parallel increase in *Cat* and *Gpx* mRNA levels. It is noteworthy that the enzymatic activity of SOD1 and the antioxidant enzymes GPx and CAT must be coordinated to efficiently remove H_2_O_2_ and avoid its accumulation. Thus, the imbalance found between the expression of SOD1 and both CAT and GPx in the hippocampus of TS mice could cause excessive accumulation of H_2_O_2_, favoring the generation of hydroxyl radicals and thereby promoting persistent OS. Importantly, the fact that other mouse models of DS, such as the Ts1Cje mouse model, do not contain *Sod1* in the trisomic segment, but have increased OS in different brain regions, suggests that other triplicated genes, including *Dscr1* or *Dyrk1A,* could also be implicated in the enhanced OS found in DS [89].

Another mechanism of defense against OS is mediated by Nrf2, a transcription factor involved in the neuronal antioxidant defense that mediates the activation of multiple antioxidant genes [90]. Under high levels of OS, Nrf2 regulates phase II antioxidant response by inducing the expression of genes that codify for many antioxidant enzymes, including NQO1, HO-1, GPx, CAT, thioredoxins, and others involved in glutathione metabolism [91]. In this study, we found that *Nrf2* was downregulated, as were the genes encoding NQO1 and HO-1 antioxidant enzymes in the hippocampus of TS mice. Thus, it is possible that this decreased *Nrf2* expression may be responsible for the inefficient induction of these antioxidant enzymes in the hippocampus of TS mice. Considering the antioxidant roles of NQO1 and HO-1 in the detoxification of quinones and heme degradation, respectively [92], the decrease in their expression could be an additional factor contributing to the increased OS in the hippocampi of TS mice. It is also possible that the decreased expression of *Nrf2* could be responsible for the lack of induction of CAT and GPx expression to compensate for the increment of SOD1 in the hippocampus of TS mice. Previous studies have demonstrated that in DS and in this mouse model the overexpression of the *Bach1* gene is associated with decreased *Nrf2* expression levels, as well as reduced phosphorylation at Ser40 with a concomitant reduction in the induction of antioxidant gene levels [12].

Collectively, an increase in hippocampal ROS production with a concomitant alteration in the antioxidant system may lead to altered redox homeostasis, exacerbating OS in the hippocampi of TS mice. This effect would further compromise neuronal homeostasis and, consequently, proper hippocampal function.

Coupled with increased OS, mitochondrial dysfunction, consistently observed in DS, also contributes to cognitive dysfunction in this condition [13,14,15]. In addition, altered mitochondrial morphology has been observed in DS fibroblasts, brain tissues, and mouse models [14,18,93,94]. Our results provide additional evidence of mitochondria with abnormal shape and size as well as ultrastructural alterations, including swelling of the matrix and focal disruption of cristae in trisomic GCs cells. These changes may indicate an accumulation of defective mitochondria in this neuronal population. Clearance of damaged mitochondria is regulated by the Pink1/Parkin pathway [95]. Pink1 accumulates on the cytosolic surface of dysfunctional mitochondria while Parkin is an E3 ubiquitin ligase, which, upon phosphorylation by Pink1, is recruited to polyubiquitinate several proteins on the surface of depolarized or damaged mitochondria, therefore signaling the organelle for degradation. Primary human fibroblasts derived from DS individuals show an accumulation of damaged mitochondria due to a deficient activation of mitophagy as a consequence of an alteration in the Pink1/Parkin signal [14,15]. Accordingly, in this study, while there is a significant increase in *Pink1* mRNA expression in the hippocampi of TS mice, supporting the existence of mitochondrial damage, *Parkin* mRNA expression is maintained at normal levels. Since Parkin initiates mitophagy, the imbalance found between *Pink1* and *Parkin* mRNA expression levels may indicate a deficiency in mitophagy in this neuronal population that leads to the accumulation of damaged mitochondria and ultimately to a high generation of ROS. In addition to anomalies in the Pink1/Parkin pathway, deficient mitophagy in the hippocampi of TS mice has also been attributed to hyperactivation of the mTOR signaling pathway, which suppresses macroautophagy [14]. In this study, we also found that the mitochondria of GCs of TS animals frequently appeared very elongated and polymorphic, suggesting an alteration of the mitochondrial dynamics of fusion/fission, which was consistent with previous observations in DS [15,16].

Brain mitochondrial bioenergetic defects are already present early in life in DS and TS pups and are enhanced across other life stages [17,18,54,94]. Thus, the ultrastructural mitochondrial abnormalities described in the GCs of the hippocampi of TS mice may be partly responsible for impaired mitochondrial function in generating the ATP required for essential neuronal processes such as the biosynthesis of proteins and neurotransmitters, synaptogenesis, axonal transport, and repair of DNA damage.

An imbalance between OS and the antioxidant system produces cell damage by the oxidation of DNA, lipids, and proteins. Regarding lipid peroxidation in the brains of TS mice, not all brain regions seem to be equally affected. While the cortex exhibits greater tolerance or better regulation against oxidative stress, the hippocampus seems to be most prone to undergoing major oxidative injury to lipids [37,38,46]. At 4, 5, and 12 months of age, the hippocampi of TS mice show high levels of different markers of lipid peroxidation such as thiobarbituric acid-reactive substances, 4-hydroxynonenal, and t8isoPGF2α [38,45,46,50]. In agreement with these studies, the results of our experiment using the BODIPY-C11 probe revealed increased lipid peroxidation in the hippocampal GCs of 6-month-old TS mice. While lipids are the biomolecules most susceptible to undergoing major oxidative injury, oxidative damage to proteins seems to require a longer period to develop in the brain of TS mice. A previous study from our laboratory demonstrated that, at 5 months of age, TS mice did not display increased levels of protein oxidation in the frontal cortex and hippocampus [38]. In addition, total levels of protein oxidation measured by total protein-bound HNE and protein nitration levels start to rise at 6 months but become significantly increased at 9 and 12 months of age in the hippocampus and frontal cortex of TS mice [49,96]. Accordingly, in this study, 6-month-old TS mice present a non-significant increment of hippocampal carbonyl levels. Thus, it is possible that the animals used in this study are still too young for an accurate assessment to be made and, therefore, that more severe oxidative damage will occur in their hippocampus as they age.

An important point for discussion is the increase in γH2AX-positive DNA damage foci found in TS GCs, even in mice of 6 months of age, before neurodegeneration and AD neuropathology appear. This can reflect elevated DNA damage or/and defective DNA repair. In mammalian neurons, DSBs are produced under physiological conditions [97], mainly as a consequence of excitotoxicity and OS, which have the potential to induce oxidative DNA damage [98,99,100]. Moreover, the transcription-associated activity of DNA topoisomerases, which produce transient DSBs, is another important source of neuronal DNA damage [101,102]. Indeed, abortive topoisomerase activity causes DNA pathogenic lesions in neurodegenerative syndromes [103,104].

Growing evidence has indicated that increased DNA damage is the basis for the aging brain and several neurodegenerative disorders, including AD [64,102,105,106,107,108]. In the case of DS, where premature brain aging and early onset of AD neuropathology occur, most studies on DNA damage have been performed in peripheral cell lines and in induced pluripotent stem cells (iPSCs) derived from DS patients [109]. Although, to the best of our knowledge, DNA damage has not been previously studied in the brain of this model of DS, satellite cells of skeletal myofibers and hematopoietic stem cells in the TS mice accumulate excessive oxidative DNA damage [110,111].

Our results demonstrated a significant increase in DSBs in TS GCs relative to control neurons. The punctate appearance of nuclear sites of DSBs corresponds to transient γH2AX-positive foci that are usually repaired and disappear within 24 h after the generation of DNA lesions, as demonstrated by previous kinetic studies on the DDR following radiation-induced DNA damage foci in neurons [97,112]. Interestingly, larger γH2AX-positive persistent DNA-damaged foci of unrepaired DNA [113] were conspicuously absent in TS GCs. This finding reasonably rules out a long-term accumulation of DNA damage at this age in TS mice. Neither have we observed a pan-nuclear γH2AX immunolabeling, which is considered unrelated to focal sites of DSBs and has been reported in response to increased neuronal activity in kainite-treated mice [114]. Moreover, the immunostaining profile of γH2AX in adult dissociated TS GCs as well as the ultrastructural analysis of their nuclei rule out the existence of significant internucleosomal DNA fragmentation with the formation of apoptotic bodies. Accordingly, several studies have reported the absence of increased apoptosis in the hippocampi of young and adult TS mice, suggesting that programmed cell death is not responsible for the hippocampal hypocellularity found in this mouse model [115,116,117,118].

In this context, it is possible that TS GCs can manage the increased DNA damage with an effective but probably slower DNA repair. In addition to the absence of persistent DNA damaged foci, we found a significant increase in mRNA levels of the replicating nucleosomal histone H2AX [72] in the hippocampi of TS mice. Previous studies have reported that limited nucleosomal histone degradation at DSBs can promote the formation of an “open” chromatin structure with enhanced mobility and access to DNA repair factors [119,120]. Therefore, the local reformation and assembly of new nucleosomes, some of them containing newly synthesized H2AX, in TS GCs could be necessary to restore the chromatin structure and ensure its original functionality after a repair has been completed. An effective DNA repair response in TS GCs was also supported by our observation of augmented expression of Ub-H2AX in TS mice hippocampi. Several studies indicated that H2AX is polyubiquitinated by the E3 ubiquitin ligases RNF8 (RING finger protein 8) and RNF168, and polyubiquitination of histone H2A variants promotes the recruitment and assembly of essential factors for DNA repair, such as p53-binding protein (53BP1), at DSB sites [105,121]. On the other hand, monoubiquitination of H2AX in polycomb repressor complexes leads to heterochromatinization and transcriptional silencing of chromatin [122], which are two physiopathological nuclear hallmarks of TS GCs [41].

A limiting factor for the efficiency of DNA repair in TS GCs may be the mitochondrial dysfunction and the consequent deficient supply of ATP for DNA repair. Indeed, DDR is a highly energy-consuming mechanism; it has been estimated that 10^4^ ATP molecules are required to repair a single DSB [105,123]. Thus, mitochondrial dysfunction potentially contributes to the increase in DNA damage foci in TS GCs.

Results from the present study indicated that the UPS functioning is disturbed in TS GCs, resulting in a reduction in both nuclear proteolytic factories (clastosomes) and the chymotrypsin-like activity of the proteasome, as well as in the accumulation of ubiquitinated proteins in cytoplasmic aggregates or inclusions. An impairment of the UPS was reported in cultured cell lines derived from DS individuals [24,30,124]. Furthermore, using redox proteomics approaches, it was demonstrated that the frontal cortex of individuals with DS presents dysregulation of proteostasis and altered profiles of ubiquitinated proteins [24,31]. In the cerebella of 18-month-old TS mice, a dysfunction of the proteasome, owing to reduced chymotrypsin-like proteolytic activity and a parallel increase in ubiquitinated proteins, was implicated in the degeneration of Purkinje cells [27]. As far as we know, this was the first study of proteasome dysfunction in the hippocampi of TS mice, a key region involved in DS cognitive deficits.

In the present study, we demonstrated for the first time in control GCs the subcellular compartmentalization of proteasomes in nuclear proteolytic factories enriched in proteasomes and ubiquitinated proteins, called clastosomes [78,80,81]. These proteolytic centers are dynamic nuclear compartments comprising molecular condensates assembled by a liquid-to-liquid phase separation mechanism [70,125]. In mammalian neurons, clastosomes seem to play an important role in regulating the nuclear concentration of short-lived protein substrates of the proteasome, such as the transcription factors c-Fos and c-Jun and ribosomal proteins, all of them essential molecular components of nuclear homeostasis [78,79,81]. Therefore, the reduction of clastosomes in TS GCs suggests a dysfunction of the UPS in controlling nuclear proteostasis. Indeed, it is thought that clastosomes contribute to the dynamic regulation of the concentration of active nuclear proteins that are proteasome substrates [78].

Another important finding of our study was the partial redistribution of Ub-protein aggregates in TS GCs consisting of switching from nuclear foci, presumably associated with clastosomes [78], to prominent cytoplasmic inclusions. The formation of cytoplasmic aggregates or inclusions containing misfolded or abnormal proteins is a neuropathological hallmark of several neurodegenerative disorders, such as AD, Huntington’s and Parkinson’s diseases, and ALS [77,126,127]. Several proteins involved in neuropathological inclusions, such as α-synuclein, tau, TDP-43, and FUS, are known to undergo an aberrant phase separation behavior that triggers protein aggregation in molecular condensates by a self-assembly process [128]. Consistent with this notion, cytoplasmic inclusions of Ub-proteins in TS GCs share certain characteristics such as the membraneless and spherical shape—defined by surface tension—of molecular condensates driven by liquid-to-liquid phase separation [128,129]. We propose that the formation of cytoplasmic inclusions of ubiquitinated proteins in TS GCs reflects a dysregulation of the neuronal UPS related to the oxidative damage of proteins [130]. The consequence is the abnormal aggregation of proteins that cannot be degraded by the proteasome. Accordingly, it is generally accepted that the ubiquitination of misfolded or abnormal proteins may lead to the elevated formation of cytoplasmic inclusions [127]. An important question is whether the formation of these inclusions is related to an early onset of the neuropathological manifestation of AD in which the proteasome fails to clear amyloid plaques or neurofibrillary tangles of tau protein [24].

The decrease in proteasome activity found in the hippocampi of TS mice does not seem to be due to alterations in the protein expression of the proteasome subunits. Proteasome activity is strongly dependent on mitochondrial-ATP production [131]. In addition, while moderate amounts of oxidative protein substrates lead to increased proteolytic activity, a reduction in proteolytic activity occurs with higher amounts of oxidative protein substrates [132]. In addition, the accumulation of β-amyloid can inhibit proteasome activity, particularly chymotrypsin-like activity [133]. At 6 months of age, the hippocampi of TS mice show an increased burden of β_1–40_ and β_1–42_ peptides [134], high levels of protein oxidation [46,49], and impaired mitochondrial bioenergetics [47,54]. Together, these neuropathological events can contribute to the decrease in the chymotrypsin-like proteasome activity (approximately 35%) reported here in the hippocampi of TS mice. The consequence is that a diminished clearance of misfolded, oxidized, or aberrant proteins can contribute to creating a toxic environment that may disturb other cellular functions in the GCs of TS mice.

A prominent cellular alteration found in TS GCs is the substantial accumulation of enlarged and polymorphic cathepsin D-positive lysosomes. A wide body of literature has demonstrated that an early neuropathological manifestation in DS and AD is the dysfunction of endosomal protein sorting and trafficking, which ultimately leads to the impairment of lysosomal functions [32,83,84]. It has been reported that fibroblasts derived from DS individuals and cultured cortical neurons from the Ts2 mouse model of DS present an enlargement of endosomal compartments as well as lysosomal dysfunctions [32,84].

Regarding the pathogenic effect of OS on lysosomal functions, previous mass spectrometry studies have reported that lipid peroxidation products such as 4-hydroxynonenal (HNE) and malondialdehyde (MDA) alter both cysteine and aspartic lysosomal proteases (cathepsins B, L, and D), resulting in lysosomal dysfunction and lipofuscinogenesis [135]. Studies on human DS brains have reported that oxidation of protein components of the autophagic–lysosomal system contributes to neurodegeneration [135]. Moreover, OS in DS cells may cause lysosomal permeabilization, leading to a translocation of the cathepsins from lysosomes to the cytoplasm, promoting its inappropriate activation [136]. Thus, the activation of cathepsin D (by increased activity and/or expression) in DS seems to be a mechanism that compensates for altered autophagy and the accumulation of toxic aggregates such as Aβ peptides [136].

Most of the lysosomes in TS GCs seem to correspond to residual lysosomal-related structures containing lipid droplets observed by electron microscopy. In this regard, it is noteworthy that lysosomes undergo marked senescence-associated alterations [137], and the formation of lysosomal residual bodies and lipofuscin granules containing lipid droplets is a characteristic feature of neuronal aging [138]. Our results with the Bodipy C11 probe revealed increased lipid peroxidation and a higher number of Bodipy C11-positive cytoplasmic bodies in the hippocampal GCs of 6-month-old TS mice. They presumably correspond to intralysosomal lipid droplets containing peroxidized lipids able to interfere with lysosomal functions. In fact, free lipid droplets in the cytosol were rarely found in TS GCs. In addition, lipid peroxidation contributes to generating several highly reactive oxidizing agents, including lipofuscin, that damage macromolecules and cellular organelles [86,138]. In this vein, the cytoplasm of the GCs from the TS mice showed an abundance of lipofuscin granules that was not found in control animals. Thus, the ultrastructural pattern of lysosomes found in TS GCs was consistent with an OS-induced dysfunction of the lysosomal system that could contribute to accelerated aging and neurodegeneration in the DS hippocampus.

Finally, both factors—increased oxidative stress and accumulation of lipofuscin found in the GCs of TS mice—can induce cellular senescence [35,86]. Accordingly, TS mice present an increased density of cells with senescent phenotypes in different areas of the hippocampus, including the granular cell layer [39,46,51].

## 5. Conclusions

In conclusion, in this study, we demonstrated that the hippocampal GCs of TS mice showed OS-associated alterations in cellular components that are essential for neuronal homeostasis, such as the DDR and proteostasis. In particular, the GCs of TS mice presented increased DNA damage and dysfunction of the UPS and lysosomal systems. The profile of the DDR and the proteostasis disturbance in TS GCs clearly correlated with the epigenetic changes and reduction in global transcriptional activity reported in our previous study in this neuronal population [41]. Collectively, the dysfunction of these neuronal processes in TS GCs may be partially responsible for the neuromorphological and functional alterations found in TS mice and in DS during early developmental stages. In addition, these alterations can promote cellular senescence, accelerated aging, and the early neuropathological manifestation of AD, and they play an important role in the progression of the hippocampal-dependent learning and memory deficits present in TS mice.

## Figures and Tables

**Figure 1 antioxidants-11-02438-f001:**
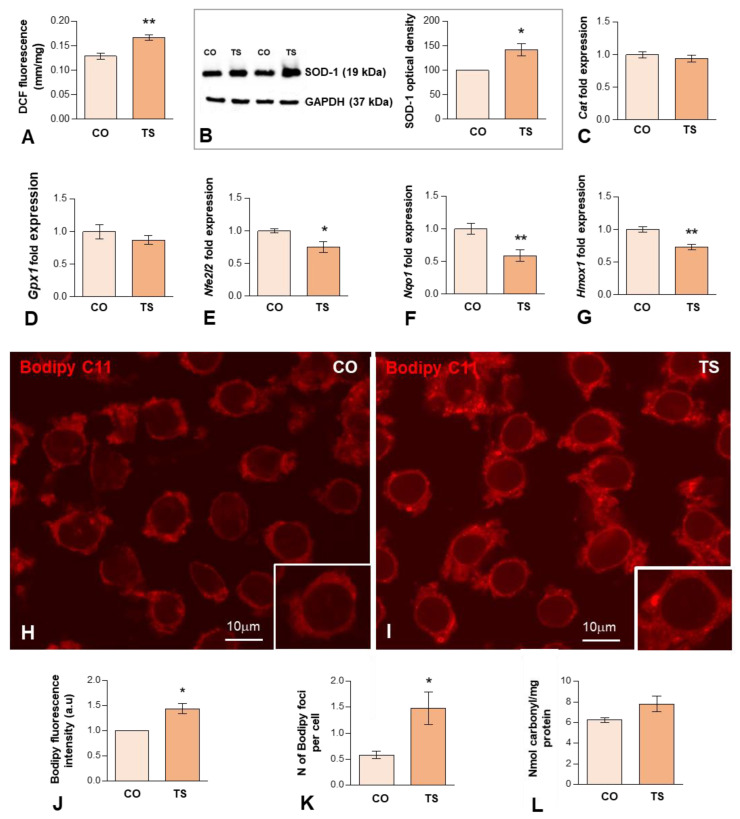
(**A**) H2DCFDA assay of hippocampal tissue homogenates, demonstrating a significant increase in hippocampal ROS levels in TS over euploid mice. Values are expressed as means ± SEMs in μM DCF fluorescence per milligram protein. (**B**) Western blot analysis of SOD1 levels showing upregulation of SOD1 protein expression in the hippocampi of TS mice relative to control (CO) animals. GAPDH was used as a loading control. (**C**,**D**) RT-qPCR determination of *Cat1* (**C**) and *Gpx1* (**D**) gene expression in hippocampal RNA extracts, showing similar levels between CO and TS mice. (**E**–**G**) RT-qPCR determination of *Nfr2* (**E**), *Nqo1* (**F**), and *Hmox1* (**G**) gene expression in hippocampal RNA extracts from animals of both genotypes, showing the downregulation of these genes in TS mice. (**H**,**I**) Representative images of Bodipy C11 fluorescent assay for peroxidized lipids in dissociated dentate gyrus GC bodies from control (**H**) and TS (**I**) mice. Note the increased fluorescent signal intensity and the presence of cytoplasmic bodies (lipid droplets) with strong Bodipy C11 signal in TS GC bodies. (**J**) Densitometric analysis revealed a significant increase in Bodipy C11 signal intensity in TS GC cytoplasm as compared with control neurons. (**K**) Quantitative analysis of the mean number of Bodipy C11-positive cytoplasmic bodies revealed a significant increase in TS GCs relative to controls. (**L**) Protein oxidative damage was determined by measuring the carbonyl content in hippocampal homogenates. Note that TS animals show higher protein oxidation than control mice. Data in (**B**–**G**,**J**,**K**) are presented as means ± SEMs of three independent experiments. *: *p* < 0.05; **: *p* < 0.01. CO = control mice; TS = Ts65Dn mice.

**Figure 2 antioxidants-11-02438-f002:**
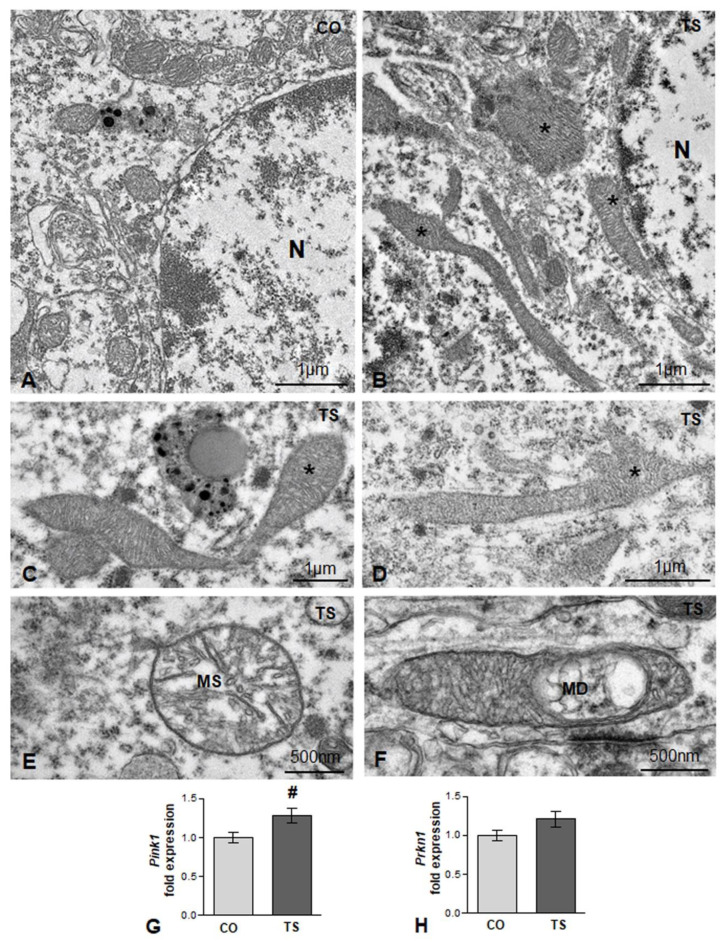
(**A**,**B**) Representative electron micrographs from control (CO) (**A**) and TS (**B**) GC bodies. Note typical small round or oval mitochondria in the control GC and the presence of larger and polymorphic mitochondria in the TS GC (asterisks). N: nucleus. (**C**,**D**) Elongated and polymorphic mitochondria in cell bodies of TS GCs (asterisks). (**E**,**F**) Detail of structural alterations in TS GC mitochondria with swelling of the matrix (MS) (**E**) and focal disruption of cristae (MD) (**F**). (**G**,**H**) RT-qPCR determination of Pink1 (**G**) and Prkn1 (**H**) gene expression in hippocampal RNA extracts from control (CO) and TS mice. Pink1 is upregulated, and a moderate but non-significant increase in Prkn1 was detected in TS hippocampi. Data in (**G**,**H**) are presented as means ± SEMs of three independent experiments. #: *p* < 0.05.

**Figure 3 antioxidants-11-02438-f003:**
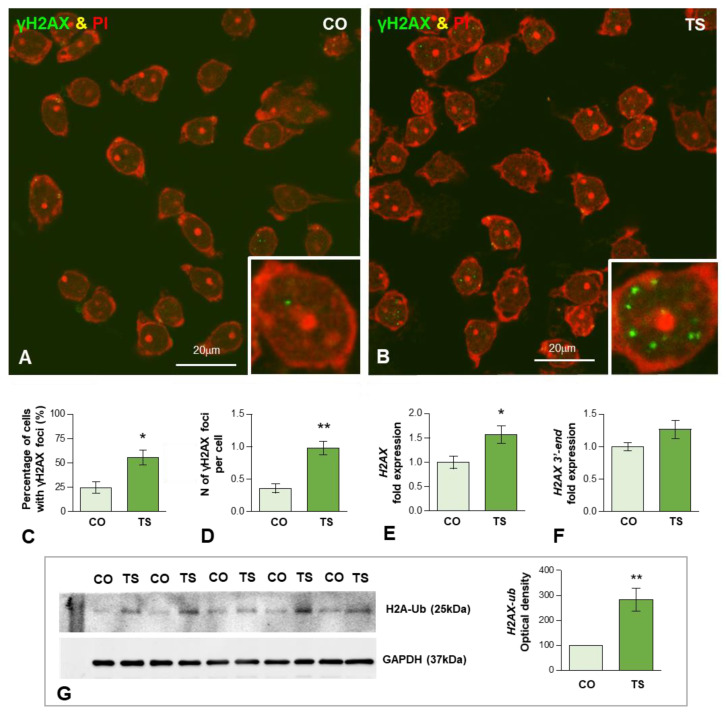
(**A**,**B**) Representative images of dissociated GCs from control (CO) (**A**) and TS (**B**) mice immunolabeled for the DNA damage marker γH2AX (green) and counterstained with propidium iodide (PI) (red) as a marker of RNA-rich structures (nucleoli and cytoplasmic regions enriched in ribosomes). Nuclear foci immunoreactive for γH2AX were preferentially observed in TS GCs ((**B**) and inset). (**C**,**D**) Quantitative analysis of the proportion of GCs containing these DNA damage foci and the mean number of γH2AX-positive nuclear foci per GC. Both parameters exhibit a significant increase in TS GCs. (**E**,**F**) RT-qPCR determination of *H2AX* mRNA (**E**) and its 3′-end unprocessed pre-mRNA (**F**) *H2AX* in hippocampal RNA extracts from control and TS mice. Whereas the expression level of mature *H2AX* mRNA significantly increased in TS hippocampi, non-significant differences were found in its 3′-end *H2AX* precursor when control and TS samples were compared. (**G**) Western blot analysis of ubiquitinated histone H2A (H2A-Ub) levels showing upregulation of H2A-Ub in the hippocampi of TS mice relative to control (CO) animals. GAPDH was used as the loading control. Data in (**C**–**G**) are presented as means ± SEMs of three independent experiments. *: *p* < 0.05, **: *p* < 0.01.

**Figure 4 antioxidants-11-02438-f004:**
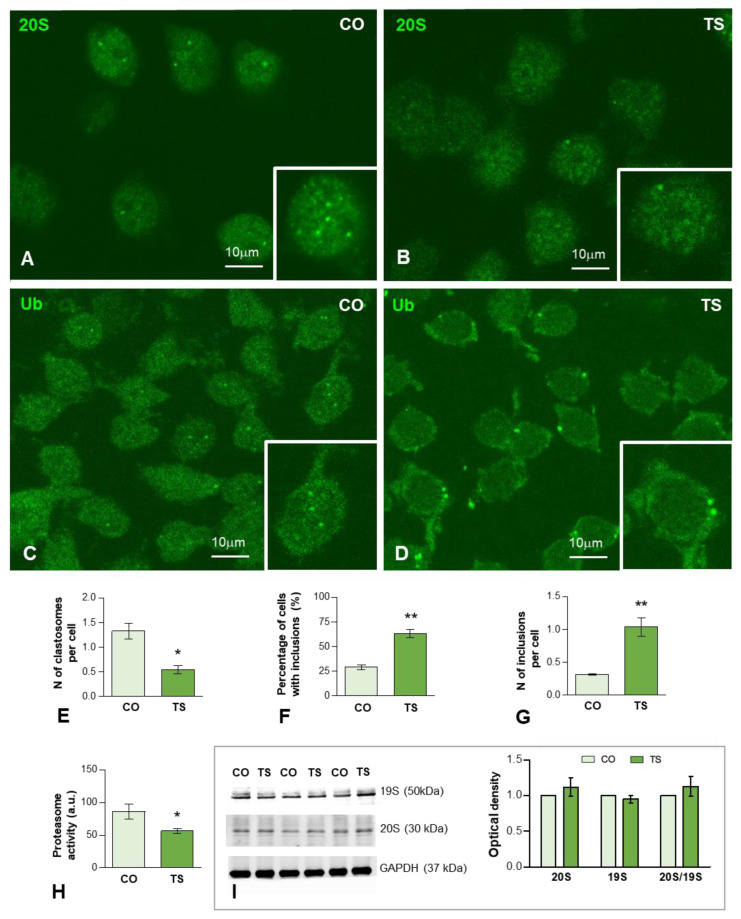
(**A**–**D**) Representative images of dissociated GCs from control (CO) (**A**,**C**) and TS (**B**,**D**) mice immunolabeled for the catalytic proteasome 20S (**A**,**B**) and ubiquitin-protein (Ub) conjugates (**C**,**D**). Nuclear foci immunoreactive for 20S proteasome and ubiquitinated proteins, identified as clastosomes, stood out against the diffuse nucleoplasmic signal and were preferentially observed in control GCs ((**A**,**B**), insets). (**E**) Quantitative analysis of the mean number of proteasome 20S-positive nuclear foci (clastosomes) per cell in control (CO) and TS GCs shows a significant increase in control relative to TS GCs. (**F**,**G**) Quantitative analysis of the proportion of GCs containing inclusions of ubiquitinated proteins (**F**) and the mean number per cell (**G**) of these cytoplasmic inclusions. Both parameters exhibit a significant increase in TS GCs. (**H**) Proteasome activity assay on hippocampal lysates reveals a decreased proteasome activity in TS hippocampi. (**I**) Western blot analysis showing protein expression of the regulatory 19S and catalytic 20S proteasome subunits in the hippocampi of control (CO) and TS mice. GAPDH was used as the loading control. Non-significant differences were found between the control and TS samples. Data of (**E**–**H**,**I**) are presented as means ± SEMs of three independent experiments. *: *p* < 0.05, **: *p* < 0.01.

**Figure 5 antioxidants-11-02438-f005:**
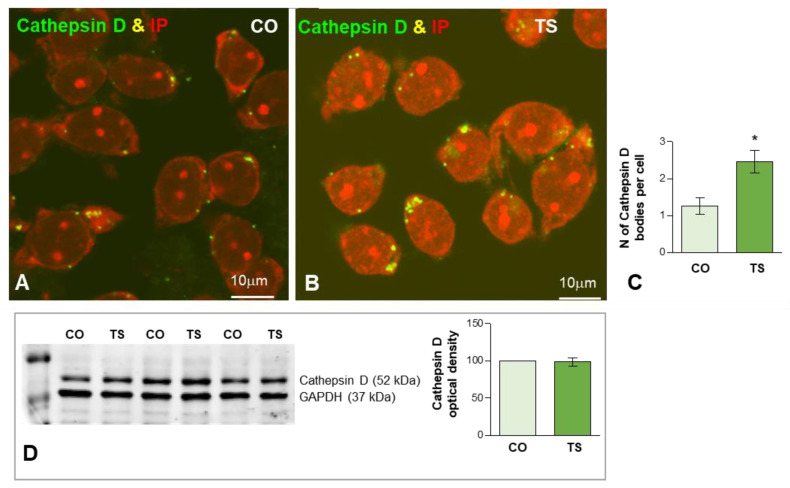
(**A**,**B**) Representative images of dissociated GC bodies from control (CO) (**A**) and TS (**B**) mice immunolabeled for cathepsin D (green) as a lysosomal marker. Note the greater abundance of cathepsin D-positive cytoplasmic bodies in TS GCs. (**C**) Quantitative analysis of the mean number of cathepsin D-positive bodies per cell reveals a significant increase in TS GCs. (**D**) Western blot analysis showing protein expression of cathepsin D in the hippocampi from control (CO) and TS mice. GAPDH was used as the loading control. Non-significant differences were found between the control and TS samples. Data of (**C**,**D**) are presented as means ± SEMs of three independent experiments. *: *p* < 0.05.

**Figure 6 antioxidants-11-02438-f006:**
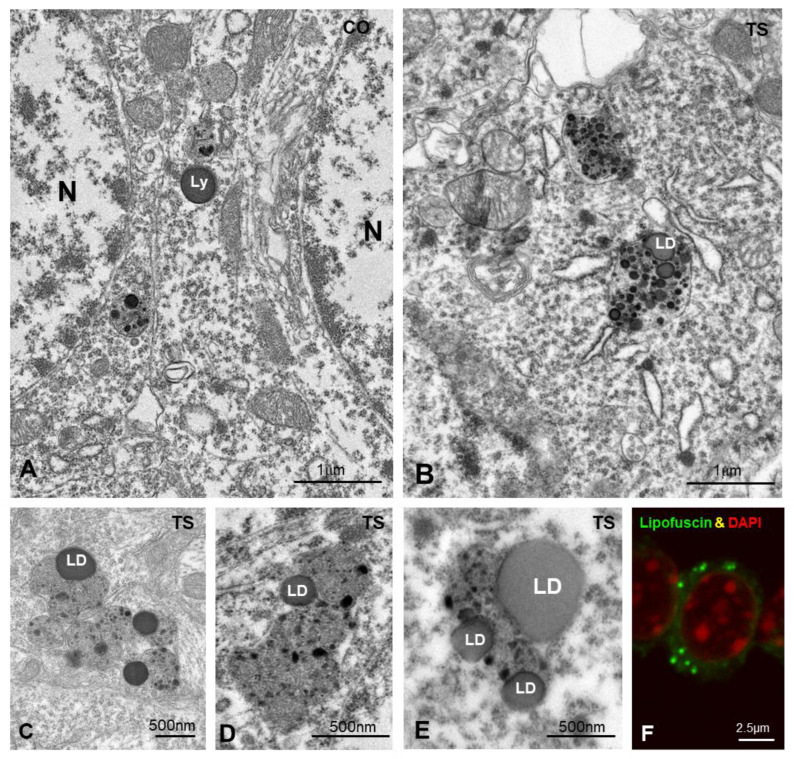
(**A**,**B**) Representative electron micrographs of control (CO) (**A**) and TS (**B**) GCs. Typical electron-dense and homogeneous lysosomes (Ly) appear in control GCs. In contrast, larger and polymorphic lysosomal-related structures containing lipid droplets (LD) were frequently found in TS GCs. (**C**–**E**) Detail of large and polymorphic lysosomal-related structures containing electron-dense granules, presumably of proteinaceous nature, and lipid droplets (LD). (**F**) Confocal microscopy of a TS GC illustrating the presence of autofluorescent lipofuscin granules (green) and counterstained with DAPI (red) for DNA.

## Data Availability

The data presented in this study are available upon reasonable request from the corresponding author.

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
