# Peer review of "Oxidative-Stress-Associated Proteostasis Disturbances and Increased DNA Damage in the Hippocampal Granule Cells of the Ts65Dn Model of Down Syndrome"

_antioxidants, 2022, doi:10.3390/antiox11122438_

Round 1

Reviewer 1 Report

Puente et al. suggest that enhancement of oxidative stress-related proteostasis and apoptosis in the hippocampal granule cells of Ts65Dn, a mouse model of DS. Many new findings support their conclusions, and the manuscript is well written. Following concerns should be addressed.

1.           Authors focused on only hippocampal granule cells of Ts65Dn mice. Are the anomalies detected in Ts65Dn mice in the present study specific in the hippocampal granule cells? Results in other regions in the hippocampus or brain should be indicated.                                                                       

2.           In figure 1, the authors just confirmed the increased expression of SOD1 in Ts65SDn. No direct evidence presented whether increased SOD1 protein level contribute enhanced oxidative stress. That’s why their conclusion, “the altered expression of SOD1 protein and of the Nrf2 gene could affect redox homeostasis, favoring the increment of OS in the hippocampi of TS mice” looks overspeculation. Furthermore, it has been already shown that enhanced oxidative stress is also detected in other mouse model of DS harboring a trisomic region excluding Sod1 gene (J Neurochem. 2009 Sep;110(6):1965-76.). The authors should discuss about this report.

Reviewer 2 Report

Oxidative-stress-associated proteostasis disturbances and increased DNA damage in the hippocampal granule cells of the Ts65Dn model of Down syndrome

The authors analyzed OS-associated alterations in hippocampal dentate granular cells using several  approaches

Introduction is well written and the aims are clearly stated. No issues.

Methods

In Section 2.1 Please identify the original commercial strain ID etc if the mice were purchased from a vendor or where the mice were sourced. Karyotyping can still reference cite #51.

Results All figures are well made and easy to interpret.

Fig 1 convincing figure

Fig 2 Fantastic EM in this figure. Stunning result

Fig 3 convincing figure

Line 377 no capital letter W in western blot

Figure 4 is convincing but has some issues. The subsets in figure 4 include 4E clastosomes, 4F intra-cell inclusions & 4G #cells with inclusions cells, but only 4E clastosomes are described in the results. The Authors need to write out the results of 4F and 4G.

Line 380, For Figure reference (Figure 4G1, G2), do the authors mean (Figure 4J) , what do the numbers represent (likely a typo)?

In Figure 5, the error bar seems to be missing for 5E (controls)

Line 420 no capital letter W in western blot

Discussion

I feel the paragraph from line 491 to 503 is interesting but doesn’t add value to the meaning pf the results. If authors delete this entire paragraph it would better focus discussion on the data in hand. Otherwise, the discussion is well thought.

Reviewer 3 Report

Review : oxidative-stress associated proteostasis disturbances and increased DNA damage in the hippocampal granule cells of the Ts65Dn model of Down syndrome.

Down syndrome (DS) caused by a trisomy of chromosome 21 (HSA21), is the most common genetic developmental disorder, with an incidence of 1 in 800 live births. DS individuals show cognitive impairment, learning and memory deficits, arrest of neurogenesis and synaptogenesis, and early onset of Alzheimer’s disease. The detailed pathogenetic mechanisms by which the extra copy of HSA21 leads to this cognitive impairment remain unknown. Oxidative stress and mitochondrial dysfunction are reported to be etiological contributors for many of the DS-related phenotype and have previously been reported to be altered in several in vitro and in vivo models of DS. Among the animal model of DS, Ts65Dn mice are widely used by scientific community: Ts65Dn mice have a partial trisomy comprising a distal portion of mouse chromosome 16 and a centromeric portion of mouse chromosome 17.

In this manuscript, the authors Puente A. et al. proposed to study the alterations associated with oxidative stress in hippocampal granule cells isolated from Ts65Dn mice (compared with control mice). More particularly, the authors were interested in this study by (i) DNA damage (ii) proteasome dysfunction and (iii) lysosomal dysfunction in hippocampal granule cells derived from Ts65Dn mice.

My general analysis of the manuscript is well presented but there are major critics.  First, there is no characterization of the cellular model used in this study (mouse hippocampal granule cells). Then, the authors did not demonstrate the presence of oxidative stress in these cells (which is normally the basis of this study). Finally, the results are not enough convincing or are too weak for supporting the conclusions of this paper. All these concerns make this manuscript not acceptable in this current form for publication in Antioxidants. This manuscript needs revisions before publication.

MATERIAL AND METHOD PART:

The procedure used for hippocampal granule cell isolation must be clearly written with details to allow the readers to perform similar studies.

RESULTS PART:

1)Major concern: The cellular model used in this study namely hippocampal granule cells isolated from brain of Ts65Dn and control mice should be clearly characterized with classical markers of such cells (immunofluorescence staining, quantitative RT-PCR analysis or other).

2) 3.1. Increased OS in the hippocampi of TS mice is associated with altered antioxidant response, 237 high levels of lipid peroxidation and mitochondrial anomalies in TS GCs

Major concern: Oxidative stress is defined as an imbalance between the production of reactive oxygen species (ROS) and the antioxidant capacity of the cell. Therefore, it is mandatory to demonstrate first that there is an imbalance between the production of ROS and the antioxidant capacity of hippocampal granule cells from Ts65Dn mice compared with the euploid counterparts. Such demonstration is not presented in Figure 1.

To demonstrate this, the authors should measure ROS production of hippocampal granule cells from Ts65Dn mice compared with the euploid counterparts (more particularly superoxide and H2O2 production). Numerous methods exist for measuring ROS produce by cells including DCFH-DA, measurement of H2O2 production by amplex red and others.

Then, measuring the activity of antioxidant enzyme or at least measuring their expression by quantitative RT-PCR will be informative to demonstrate for instance a reduced antioxidant capacity in hippocampal granule cells from Ts65Dn mice.

Then the enhanced H2O2 production by the hippocampal granule cells can be potentially attributable to the elevated level of SOD1 protein (encoded by a HSA21 gene) in these cells, which converts superoxide into H2O2.

Also, the author investigated lipid peroxidation because of oxidative stress with Bodipy C11 probe but did not investigate protein oxidation which seems more adequate considering the general aim of this study “proteostasis and proteasome dysfunction”.

Minor concern:

Fig 1C. Quantitative RT-PCR of Nrf2 gene is not informative, given that the authors do not investigate more this pathway such as the expression of the target genes that are activated or repressed by Nrf2 pathway.

Fig 1A. the quality of the images is weak given that the background of these images is not black (but also red). Also, the cells should be stained by a nuclear marker (DAPI, Hoechst or other).

As for the image for TS granule cells in which the authors have made a zoom of one granule cell to see the Bodipy C11 foci, this should be done for the control granule cells

Page 7 and Figure 2: the authors explored the structural mitochondrial alterations in TS GCs compared to control GCs by electron microscopy, given that oxidative stress can lead to mitochondrial structural alterations and/or dysfunction. Instead of investigating Pink1 Parkin pathway (just by the expression of Pink1 and Parkin by quantitative RT-PCR, which is weak), the authors should confirm that these structural mitochondrial alterations in TS GCs are also associated with mitochondrial dysfunction by measuring for example mitochondrial potential ΔΨm or by measuring mitochondrial respiration.

3)3.2. Oxidative stress in TS GCs induces DNA damage

Minor concern: Figure 3

Figure 3D should be before Figure 3C to improve the understanding of Fig 3C. The Y-axis of Fig 3D should be corrected as “Percentage of cells with γH2AX foci (%)”

As for the image for TS granule cells in which the authors have made a zoom of one granule cell to see the γH2AX foci, this should be done for the control granule cells

4)3.3. Increased OS is associated with proteostasis disturbances in TS GCs

3.3.1. Dysfunction of the proteasome

The demonstration of proteasome dysfunction in Ts65Dn granule cells is weak, given that (i) only chymotrypsine like activity has been measured (trypsine like and caspase like activity should be measured) (ii) no difference in 20S protein level by western blotting is found (iii) the conclusions that emerge from Figure 1A 1B are not convincing

Minor concern: Fig 4A.B

The granule cells should be stained by a nuclear marker (DAPI, Hoechst or other). As for the image for control granule cells in which the authors have made a zoom of one granule cell to see the number of Ub inclusions, this should be done for the TS granule cells

Fig 4AB and 4E. The quantitative analysis is not representative of the images seen in Fig 4A and 4B. The number of 20S foci is similar between control and TS condition in Fig4A and 4B whereas in Fig 4E, the authors showed a greater number of 20S in control granule cells compared to TS granule cells.

The average number of clastosomes per cell is not 1,4 when we are analysing Fig4A.

Fig 4 C.D the quality of the images is weak given that the background of these images is not black (but also green). Also, the cells should be stained by a nuclear marker (DAPI, Hoechst or other).

Figure 4G should be before Figure 4F to improve the understanding of Fig 4F. The Y-axis of Fig 3D should be corrected as “Percentage of cells with inclusions (%)”

As for the image for control granule cells in which the authors have made a zoom of one granule cell to see the number of Ub inclusions, this should be done for the TS granule cells

5) 3.3.2. Dysfunction of lysosomal system

Minor concern: Fig 5A, B and C

Fig 5A and 5B with Histogram 5C. The representative images Fig 5A and 5B, and the results of Fig 5C in which the authors showed a ~2-fold increase of the number are contradictory. Indeed the number of cathepsin bodies per TS granule cells (Fig5B) is much more than 2,..(as seen in Fig 2C).

Round 2

Reviewer 1 Report

The manuscript has been improved.